# Mid-Cretaceous marine Os isotope evidence for heterogeneous cause of oceanic anoxic events

Hironao Matsumoto[1✉], Rodolfo Coccioni [2], Fabrizio Frontalini [3], Kotaro Shirai [1], Luigi Jovane [4], Ricardo Trindade [5], Jairo F. Savian [6] & Junichiro Kuroda[1]

During the mid-Cretaceous, the Earth experienced several environmental perturbations, including an extremely warm climate and Oceanic Anoxic Events (OAEs). Submarine volcanic episodes associated with formation of large igneous provinces (LIPs) may have triggered these perturbations. The osmium isotopic ratio ($^{187}Os/^{188}Os$) is a suitable proxy for tracing hydrothermal activity associated with the LIPs formation, but $^{187}Os/^{188}Os$ data from the mid-Cretaceous are limited to short time intervals. Here we provide a continuous high-resolution marine $^{187}Os/^{188}Os$ record covering all mid-Cretaceous OAEs. Several OAEs (OAE1a, Wezel and Fallot events, and OAE2) correspond to unradiogenic $^{187}Os/^{188}Os$ shifts, suggesting that they were triggered by massive submarine volcanic episodes. However, minor OAEs (OAE1c and OAE1d), which do not show pronounced unradiogenic $^{187}Os/^{188}Os$ shifts, were likely caused by enhanced monsoonal activity. Because the subaerial LIPs volcanic episodes and Circum-Pacific volcanism correspond to the highest temperature and $pCO_2$ during the mid-Cretaceous, they may have caused the hot mid-Cretaceous climate.

[1] Atmosphere and Ocean Research Institute, The University of Tokyo, 5-1-5 Kashiwanoha, Kashiwa 277-8564, Japan. [2] University of Urbino, Carlo Bo, 61029 Urbino, Italy. [3] DiSPeA, University of Urbino Carlo Bo, Campus Scientifico Enrico Mattei, Località Crocicchia, 61029 Urbino, Italy. [4] Instituto Oceanográfico, Universidade de São Paulo, Praça do Oceanográfico, 191, São Paulo, SP 05508-120, Brazil. [5] Instituto de Astronomia, Geofísica e Ciências Atmosféricas, Universidade de São Paulo, Rua do Matão, 1226, São Paulo, SP 05508-090, Brazil. [6] Departamento de Geologia, Instituto de Geociências, Universidade Federal do Rio Grande do Sul, Avenida Bento Gonçalves, 9500, Porto Alegre, RS 91501-970, Brazil. ✉email: matsumoto@aori.u-toyko.ac.jp

The mid-Cretaceous (late Barremian to Turonian: ~121–90 Ma) is commonly regarded as one of the extremely warm geological intervals of the Phanerozoic Eon[1]. The oxygen isotopic ratio of carbonate ($\delta^{18}O_{carb}$) and $TEX_{86}$-Sea Surface Temperature (SST) proxies have revealed that the Cenomanian to Turonian climate was much warmer than that of today[1–3]. In addition, palaeobotanical[4] and palaeontological data from the Late Cretaceous (Turonian to Coniacian)[5] suggest that a warm climate prevailed in the Arctic region. The warm climate during the mid-Cretaceous is considered to have been sustained by a high $pCO_2$ (e.g., ~1500 ppmv during the Cenomanian[6]) derived from active outgassing associated with the production of oceanic crust and/or massive volcanic activity[7].

This greenhouse world experienced distinctive repeated oceanic anoxic events (OAEs), representing major perturbations in the carbon cycle characterized by deposition of organic-rich sediments in various depositional settings. Many organic-rich lithological intervals have been reported from the mid-Cretaceous Tethyan sedimentary record[8,9]. The early Aptian OAE1a and latest-Cenomanian OAE2, the most prominent mid-Cretaceous OAEs, were typified by worldwide deposition of thick organic-rich horizons[10] (e.g., 1–2 m thick at the Umbria–Marche Basin). Additionally, other minor OAEs (e.g., OAE1b, OAE1c, and OAE1d), which have been reported mainly from the Tethys and Atlantic Oceans[9,11–13] and a part of the Pacific region[14], are regarded as regional to supra-regional marine anoxic events. As mid-Cretaceous OAEs were often accompanied by intensive marine biotic crises[15], understanding the factors that triggered the OAEs is important for unraveling the evolution of the Cretaceous marine ecosystem.

Massive volcanic events associated with the formation of large basaltic plateaus called large igneous provinces (LIPs) are the most probable triggering factors of environmental perturbations[15]. Because the radiometric ages of the basaltic plateaus correspond to the sedimentary ages of major OAEs and the species turnovers of marine calcareous plankton, these events are thought to have been linked[15–17]. The Os isotopic variations ($^{187}Os/^{188}Os$) in the sedimentary record further support a causal linkage between the LIPs volcanism and the onset of marine environmental perturbations. The $^{187}Os/^{188}Os$ values of seawater represent the balance between radiogenic material from a continental source (~1.0–1.5) and unradiogenic material from hydrothermal activity, weathering of mafic rocks, and extraterrestrial materials (~0.12)[18]. During the mid-Cretaceous OAEs (e.g., OAE1a and OAE2), the Os isotopic ratios of the sedimentary record show highly unradiogenic values (~0.2), which have been interpreted to indicate massive input of unradiogenic Os associated with LIPs formation[19–21]. However, the Os isotopic record from the mid-Cretaceous is limited to the latest Barremian to early Albian and Cenomanian–Turonian transitional intervals[13,16,19–21], and these records are not sufficiently long to elucidate the evolution of the prolonged hydrothermal activity associated with volcanic episodes during the mid-Cretaceous.

Here, we reconstruct a continuous marine Os isotopic record from the middle Albian to the uppermost Cenomanian using a pelagic sedimentary record from the Umbria–Marche Basin (central Italy) and the borehole core from Ocean Drilling Program (ODP) Site 763 (Exmouth Plateau, southeast Indian Ocean) (Fig. 1). Integrating our data with previously published information, we provide a continuous Os isotopic record from the late Barremian to early Turonian, covering all mid-Cretaceous OAEs, and discuss the long-term hydrothermal record of the mid-Cretaceous. As a result, we found that hydrothermal activity associated with the formation of LIPs was enhanced during the mid-Aptian, late Albian, and end-Cenomanian. In addition, temporal intensification of continental weathering was observed during the early Albian, which may be caused by temporal global warming. From the Os isotopic variations, we found that mid-Cretaceous OAEs can be classified into two types according to their triggering factors as: (1) volcanic-induced OAEs (e.g., OAE1a, Wezel Level, Fallot Level, and OAE2) with unradiogenic Os isotopic shifts; and (2) monsoon-induced OAEs (OAE1c and OAE1d) without unradiogenic Os isotopic shifts. Besides, the warmest interval during the mid-Cretaceous corresponded to a phase of enhanced subaerial volcanic episodes with no evidence of long-term enhanced hydrothermal activity. Thus, we conclude that subaerial volcanic episodes and the subsequent outgassing were the main cause of the warm mid-Cretaceous climate.

## Results

Limestone, marlstone, mudstone, and black shale samples ranging from the middle Albian to the upper Cenomanian were collected from the PLG core[8] (43°32′42.72″N, 12°32′40.92″E) and the Bottaccione section[9] (43°21′56.04″N, 12°34′57.56″E) in the Umbria–Marche Basin (Central Italy). The Umbria–Marche sedimentary record comprises pelagic sedimentary facies of the Tethyan Ocean (Fig. 1) and is characterized by a lack of coarse terrigenous materials. The PLG core is a borehole core drilled near the PLG section[12] that covers the uppermost Barremian to the lowest Cenomanian. The Bottaccione section is a pelagic sedimentary sequence located in the same basin that includes, for the Cretaceous, the uppermost Albian to the Maastrichtian[9].

Using lithostratigraphy, biostratigraphy, and carbon and osmium isotopic stratigraphy[8,9,22], we reconstruct a continuous composite stratigraphic record of the Umbria–Marche Basin during the mid-Cretaceous (Fig. 2). The upper Barremian of the composite record belongs to the Maiolica Formation and consists of white cherty limestone cyclically intercalated with thin (~few centimeters) black shale layers[8]. The ~2 m-thick organic-rich interval, known as the Selli Level[23] occurs around the Barremian–Aptian boundary and records the regional sedimentary expression of OAE1a. Above the Selli Level, the Aptian sedimentary record consists of marly limestone with some black shale layers (e.g., the Wezel and Fallot Levels[13]) belonging to the Marne a Fucoidi Formation[8]. The Fallot and Wezel Levels are only reported from the Tethyan Region. However, their accurate extent has not been constrained so far because of the limited geological research focusing on their equivalent intervals outside the Tethyan region. Several pronounced organic-rich intervals (i.e., Jacob, Kilian, Urbino/Paquier, and Leenhardt Levels), collectively called OAE1b, appear around the Aptian–Albian boundary[8,12]. The Albian sediments consist of mudstones intercalated with cyclic thin (~few centimeters) black shale layers[8,12,24]. A peculiar ~2-m-thick interval in the upper Albian, called the Amadeus Segment[24], is located in the middle part of OAE1c that spans almost the entire *Biticinella breggiensis* planktonic foraminiferal Zone[8,25]. The muddy interval ends in the upper Albian, and the lithology changes to the white and reddish limestones of the Scaglia Bianca Formation[9]. The last cyclic upper Albian organic-rich layers are known as the Pialli Level[26], which represents the regional sedimentary expression of OAE1d[9]. At the end-Cenomanian, a thick organic-rich interval, known as the Bonarelli Level, is the regional sedimentary expression of OAE2[9]. Osmium and carbon isotopic records from the upper Barremian to lower Albian in the Umbria–Marche Basin have already been reported[12,13,16,19,27–29].

Sedimentary rock samples were also collected from ODP Site 763B in the western part of the central Exmouth Plateau (20°35.21′S, 112°12.51′E, northwestern Australian margin, subtropical Indian Ocean) at 1368 m below the sea surface[30]. The sediments were deposited at upper bathyal depth (~200–500 m

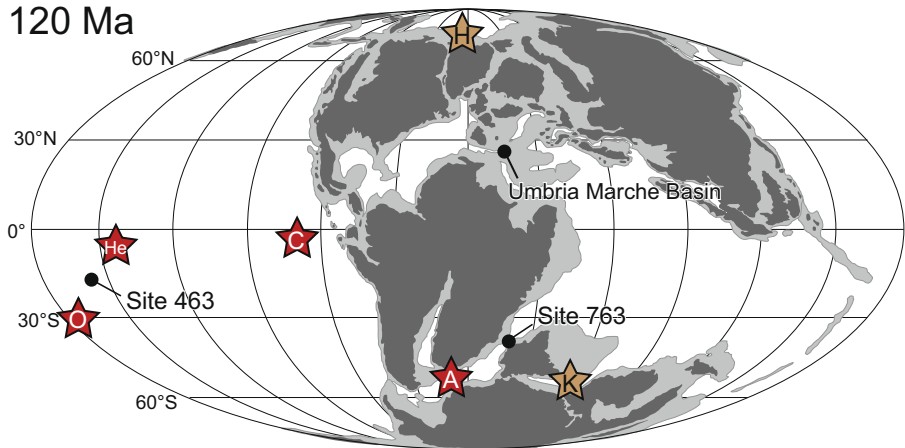

**Fig. 1 Paleogeographic reconstruction (120 Ma)[16].** A: Agulhas Plateau, C: Caribbean Plateau, K: Kerguelen Plateau, H: High Arctic Large Igneous Provinces, He: Hess Rise, O: Ontong Java Nui.

**Fig. 2 $^{187}O/^{188}Os_i$ and $\delta^{13}C_{carb}$ records of the Umbria–Marche Basin and ODP Site 763B. a** Umbria–Marche Basin: lithology, biostratigraphy, biostratigraphy, and geochemical data are from Coccioni et al.[8,12], Coccioni and Premoli Silva[9], Coccioni[65], Turgeon and Creaser;[19] Tejada et al.[20], Li et al.[27], Savian et al.[28], Matsumoto et al.[13,16], Percival et al.[29], and this study. **b** ODP Site 763B: lithology, biostratigraphy, and geochemical data are from Haq et al.[30], Bralower et al.[35], and this study. Ma. Maiolica, Sc. Ro. Scaglia Rossa, Ba. Barremian, Tur. Turonian, *G. Globigerinelloides, apt aptiensis, L. Leupoldina, fer. ferreolensis, algeria. algerianus, H. Hedbergella, troc. trocoidea, M. Microhedbergella, P. Paraticinella, mini. miniglobularis, reni. renilaevis, T. Ticinella, B. Biticinella, Ps. Pseudothalmanninella, su. subticinensis, ticine. ticinensis, Pth. Parathalmanninella, appen. appenninica, Th. Thalmanninella, R. Rotalipora, r. reicheli, g. globotruncanoides, W. Whiteinella, a. archaeocretacea, Hv. Helvetoglobotruncana, h. Helvetica, J Jacob, K Kilian, U Urbino, L Leenhardt.*

below sea level[30]) and consisted of calcareous claystone to clayey nannofossil chalk. The Aptian–Albian boundary is estimated to lie between cores 37 and 36, but the exact position is uncertain (Fig. 2b)[31]. The Cenomanian–Turonian boundary falls in the gap between cores 23 and 22 (Fig. 2b). We collected sedimentary rock samples from ODP Site 763B cores 36 to 21, which cover the middle Albian to lowest Turonian.

The $\delta^{13}C_{carb}$ record of sedimentary rock samples from the Umbria–Marche Basin (PLG core and Bottaccione section) shows a gradual decline from ~3‰ in the middle Albian (~40 m: Fig. 2) to 2.1‰ in the Pialli Level (~90 m: Fig. 2, Supplementary Tables 1 and 2). The $\delta^{13}C_{carb}$ curve shows a slight positive excursion (~0.4‰: Fig. 2) within the Pialli Level (~90 m). Above this level, $\delta^{13}C_{carb}$ shows a gradual positive shift from 2‰ (~110 m) to 3‰ toward the end of Cenomanian (~130 m) (Fig. 2a). Our data are concordant with those of the previous studies[9,32]. At ODP Site 763B, $\delta^{13}C_{carb}$ shows a gradual negative shift from ~3‰ at the Aptian–Albian boundary (~520 meters below the sea level [mbsf]) to 1‰ around the Albian–Cenomanian boundary (~434 mbsf) (Fig. 2b). The $\delta^{13}C_{carb}$ values of some core samples are highly negative (−2‰ to 0.5‰), which might indicate some diagenetic alteration possibly caused by remineralization of organic matter. The $\delta^{13}C_{carb}$ values increase during the Cenomanian (from 420 to 410 mbsf), which are concordant with the Tethyan sedimentary record (Fig. 2b). The $\delta^{13}C_{carb}$ data exhibit a large negative shift from ~2‰ at 410 mbsf to −3‰ at the Cenomanian–Turonian boundary (~380 mbsf) (Fig. 2b). Albian–Cenomanian sedimentary rock samples from ODP Site 762 C (western part of the Exmouth Plateau) also show highly negative $\delta^{13}C_{carb}$ values, which are interpreted to indicate diagenetic alteration[33]. Although the exact mechanism is unclear at present, it is likely that the carbon isotopic records at ODP Sites 763B and 762 C have experienced a similar diagenetic overprint around the Cenomanian–Turonian boundary.

In the Umbria–Marche sedimentary record (PLG core and Bottaccione section), $^{187}Os/^{188}Os_i$ varies from 0.37 to 0.75, except for one sample (BTT450) with an extraordinarily high value (~1.1) and a relatively high $^{187}Re/^{188}Os$ ratio (~10) (Supplementary Tables 3 and 4). Because the Re-Os information of outcrop samples with high Re/Os values can be easily altered by weathering[16], we considered this point as an outlier and excluded it from our discussions. The $^{187}Os/^{188}Os_i$ values of ODP Site 763B span from 0.47 to 0.75; this range is concordant with the Tethyan sedimentary record (Figs. 2 and 3).

## Discussion

**Mid-Cretaceous Os isotopic fluctuations**. The composite mid-Cretaceous Os isotopic data reveal that pronounced unradiogenic shifts (i.e., lower values) occurred during the early to mid-Aptian, late Albian, and end-Cenomanian, and radiogenic shifts took place at the Aptian–Albian boundary and during the Cenomanian (Figs. 2 and 3a). As marine Os isotopic ratios ($^{187}Os/^{188}Os$) represent the balance between unradiogenic Os input (mantle and extraterrestrial material) and radiogenic Os input (continental material), these Os isotopic variations reflect changes in the Os fluxes from these sources.

In the uppermost Barremian, $^{187}Os/^{188}Os_i$ exhibits relatively radiogenic values (~0.6–0.7) in the Umbria–Marche Basin (Gorgo a Cerbara section and PLG core) and the Pacific record (Deep Sea Drilling Project [DSDP] Site 463) (Figs. 2 and 3a)[20,21,29]. On the contrary, $^{187}Os/^{188}Os_i$ shows sharp drops to ~0.2–0.36 in the lower to mid-Aptian black shales, namely the Selli, Wezel, and Fallot Levels (Figs. 2 and 3a)[13,20,21,29]. As the sedimentary ages of these unradiogenic Os isotopic shifts correspond to the

radiometric ages of the Ontong Java, Manihiki, and Hikurangi Plateaus, which once formed a single large oceanic plateau called Ontong Java Nui (OJN) (Fig. 3a, e), these unradiogenic Os isotopic shifts were likely triggered by a massive input of mantle-derived unradiogenic Os through hydrothermal activity and warm- and low-temperature submarine weathering at OJN[13,20,21,34]. This possibility is further supported by $^{87}Sr/^{86}Sr$ and sulfur isotopic evidence (Fig. 3b, c). Hydrothermal fluid is characterized by unradiogenic $^{87}Sr/^{86}Sr$ and more negative $\delta^{34}S$ values than those of seawater; thus, the unradiogenic $^{87}Sr/^{86}Sr$ values (e.g., Fig. 3b)[35] and negative sulfur isotopic ($\delta^{34}S_{barite}$) excursion (Fig. 3c)[36] during the early to mid-Aptian further support the hypothesis of enhanced hydrothermal activity and of warm- and low-temperature submarine weathering.

Lechler et al.[37] suggested that, in addition to hydrothermal activity, subaerial weathering of OJN basalt could have played a significant role in causing large unradiogenic shifts. However, modeling studies have indicated that weathering of a huge amount (~30–60%) of the Ontong Java Plateau would have been required to explain such large unradiogenic Os isotopic shifts[20]. Given that most of the plateau was emplaced under submarine conditions[38], we infer that hydrothermal activity was likely the major cause of early to mid-Aptian unradiogenic Os isotopic shifts.

During OAE1b, the $^{187}Os/^{188}Os_i$ values of Tethyan and Pacific sedimentary records show a radiogenic shift from 0.5 to 0.7 (Fig. 2)[16]. Our $^{187}Os/^{188}Os_i$ data from ODP Site 763B also reveal that radiogenic $^{187}Os/^{188}Os_i$ values prevailed during the early Albian (Figs. 2 and 3a). The radiogenic shift of $^{187}Os/^{188}Os_i$ corresponds to the $^{40}Ar/^{39}Ar$ ages of the Kerguelen Plateau basalt (Fig. 3)[39–41] and an increase in temperature, as indicated by $\delta^{18}O_{carb}$ of belemnites, the $TEX_{86}$-SST index, and the demise of glendonite (a pseudomorph after ikaite that is a hydrated calcium carbonate formed under low-temperature conditions) in the Arctic region (Fig. 3a, d, e)[42–44]. Thus, the radiogenic Os isotopic shift during OAE1b has been interpreted as indicating enhanced continental weathering triggered by global warming caused by outgassing from volcanic episodes at the Kerguelen Plateau[16]. As most of the Kerguelen Plateau was emplaced subaerially at a high latitude in the Indian Ocean, unradiogenic Os inputs from hydrothermal activity and weathering of the basaltic rock were insignificant and did not influence the marine Os isotopic record[16]. In addition, $^{87}Sr/^{86}Sr$ values show a radiogenic shift during the early Albian[35], which also supports enhanced continental weathering during OAE1b[16].

The $^{187}Os/^{188}Os_i$ data exhibit two pronounced unradiogenic shifts during OAE1c (Figs. 2 and 3a). These excursions, which were observed in both the Tethyan realm (Umbria–Marche Basin) and in the Indian realm (ODP Site 763B) (Figs. 2 and 3), can be ascribed to a decrease in the radiogenic Os input through continental weathering or an increase in the unradiogenic Os input from the mantle or extraterrestrial material. To explain the unradiogenic shifts solely by the decreased continental weathering, a rapid decrease in temperature is required; however, no data support this possibility. Therefore, the two unradiogenic shifts may represent an increase in the input of unradiogenic Os. One of the possible sources of unradiogenic Os is extraterrestrial materials, but the unradiogenic shifts during OAE1c are much longer and smaller (~1 Myr and $^{187}Os/^{188}Os$ ~0.4) than those during massive meteorite impacts (~200 kyr, $^{187}Os/^{188}Os$~0.1–0.2)[45]. In addition, subaerial basaltic eruptions at low latitudes have not been reported during OAE1c. Thus, the most probable candidate for the unradiogenic shifts during OAE1c is an increase in hydrothermal activity. Indeed, $\delta^{34}S_{barite}$ data show negative values during OAE1c (Fig. 3c), which implies

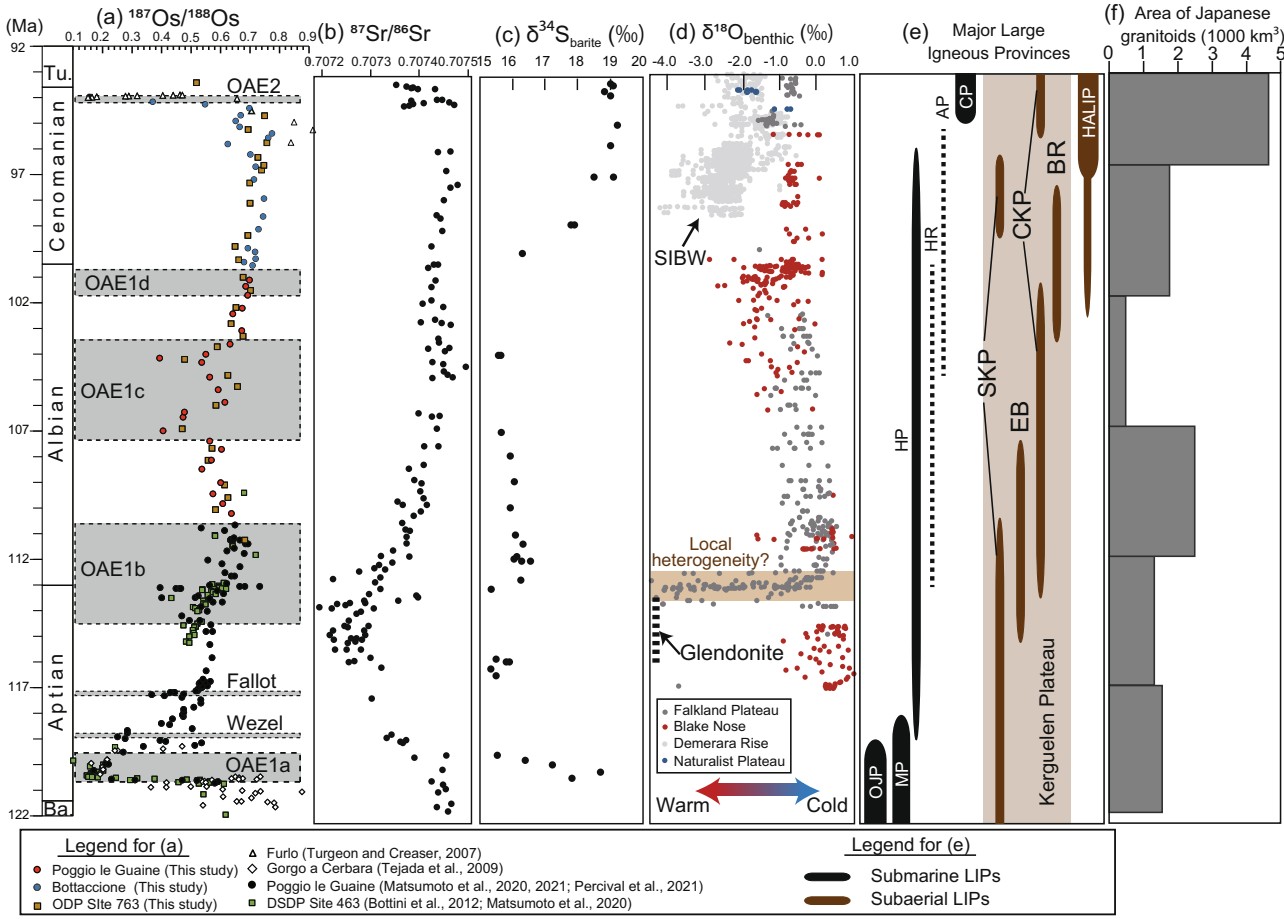

**Fig. 3 Compilation of $^{187}$Os/$^{188}$Os$_i$ and other geochemical data. a** $^{187}$Os/$^{188}$Os$_i$ from Turgeon and Creaser[19], Tejada et al.[20], Bottini et al.[21], Matsumoto et al.[13,16], Percival et al.[29], and this study. **b** $^{87}$Sr/$^{86}$Sr from Bralower et al.[35]; **c** $\delta^{34}$S$_{barite}$ from Paytan et al.[34]; and **d** $\delta^{18}$O$_{carb}$ from Fassel and Bralower[66], Huber et al.[1,11,67], Gustafsson et al.[68], Moriya et al.[69], Petrizzo et al.[70], and Friedrich et al.[2]. SIBW: the values of warm, saline intermediate to bottom waters. Age-scale is from Gale et al.[71]. Abbreviations and age sources in **e**: OJP Ontong Java Plateau (Mahoney et al.[72] and Tejada et al.[73]), MP Manihiki Plateau (Ingle et al.[74] and Timm et al.[75]), HP Hikurangi Plateau (Hoernle et al.[46]), HR Hess Rise (Vallier et al.[47]), AP Agulhas Plateau (Parsiegla et al.[48]), CP Caribbean Plateau (Sinton et al.[51]), SKP Southern Kerguelen Plateau, EB Elan Bank, CKP Central Kerguelen Plateau, BR Broken Ridge (Coffin et al.[39], Duncan et al.[40], and Jiang et al.[41]), HALIP High Arctic Large Igneous Provinces (Naber et al.[61]). Dashed lines in **e**: ages estimated from plate-tectonic reconstructions. **f** Japanese granitoid ages from Takagi[62].

hydrothermal sulfur input. However, the $^{87}$Sr/$^{86}$Sr values do not show a significant unradiogenic shift, possibly because of the long residence time of Sr (~3 Myr) compared to that of Os (20–50 kyr[18]) and the volatile feature of the highly oxidized form of OsO$_4$.

The estimated ages of the Hikurangi Plateau[46], Kerguelen Plateau[39–41], Hess Rise[47], and Agulhas Plateau[48] cover the sedimentary ages of OAE1c (Fig. 3). Thus, the hydrothermal activity associated with the formation of these oceanic plateaus may have triggered the unradiogenic Os isotopic shifts. However, the age constraints on the Hikurangi Plateau, Hess Rise, and Agulhas Plateau are too poor to conclude when the hydrothermal activity occurred, and further studies are required to determine the exact source of these unradiogenic Os shifts.

After OAE1c, the $^{187}$Os/$^{188}$Os$_i$ values gradually shift to be more radiogenic ~0.7 (Fig. 3a), which may reflect the weakening of hydrothermal activity. This possibility is supported by the sulfur isotopic ratio and strontium isotopic evidence: $\delta^{34}$S$_{barite}$ shows a positive excursion during the late Albian (Fig. 3c) that can be interpreted as a decrease in the hydrothermal sulfur input with low $\delta^{34}$S[49]. The positive excursion of $\delta^{34}$S$_{barite}$ can be also explained by an increase in sulfur reduction during the early Cenomanian. However, considering the organic-rich sediments

are more pronounced during Albian than Cenomanian at the Umbria–Marche Basin, sulfate reduction should also have been more significant during the Albian than Cenomanian. Therefore, we consider that the decrease in the volcanic sulfur is a more important factor for explaining the positive $\delta^{34}$S excursion than the sulfate reduction. The positive excursion of $\delta^{34}$S$_{barite}$ during the Cenomanian postdates the cessation of the Os isotopic fluctuations (Fig. 3a, c). Since the residence time of sulfur in the ocean is longer than Os, the onset of the changes of $\delta^{34}$S$_{barite}$ could have been more gradual and possibly postdated the radiogenic Os isotopic shift. In addition, $^{87}$Sr/$^{86}$Sr show radiogenic values (Fig. 3b) that also support the weakening of the input of hydrothermal unradiogenic Sr[35]. The $^{187}$Os/$^{188}$Os values do not show any significant fluctuation during OAE1d, which likely suggests the absence of intensive submarine volcanism.

The $\delta^{18}$O$_{carb}$ values of benthic and planktonic foraminifera suggest a temperature increase during the Cenomanian, with the warmest conditions recorded from the Cenomanian to Turonian[1,2]. The enhanced chemical weathering caused by the warm climate may have accelerated and intensified inputs in radiogenic continental Os, which may also have contributed to the radiogenic $^{187}$Os/$^{188}$Os$_i$ shifts after OAE1c (Fig. 3).

Stable radiogenic $^{187}Os/^{188}Os_i$ values during the Cenomanian (~0.7) were followed by a sudden drop just below the OAE2 interval (Fig. 3)[19,50]. As the sedimentary ages of the unradiogenic shifts correspond to the $^{40}Ar/^{39}Ar$ ages of the Caribbean Plateau[51], the unradiogenic shifts can be explained by an increase in the unradiogenic Os input associated with emplacement of the submarine basaltic plateau. This possibility is further supported by the unradiogenic shift of $^{87}Sr/^{86}Sr$ after OAE2[35]. Although the weathering of the basaltic plateau may have contributed to the unradiogenic Os isotopic shifts, geological evidence of how much of the plateau was exposed subaerially is still lacking, precluding further discussion of this possibility. $\delta^{34}S_{barite}$ data around OAE2 are scarce, but $\delta^{34}S$ of pyrite and carbonate-associated sulfates (CAS) around OAE2[52] has been intensively investigated instead. $\delta^{34}S_{CAS}$ and $\delta^{34}S_{pyrite}$ showed a positive excursion (2–4‰) across the OAE2, suggesting an enhanced sulfate reduction[52]. Considering the global oceanic anoxia and short duration of unradiogenic Os isotopic shift during OAE2 (~600 kyr), the effect of the sulfate reduction could have overwhelmed the effect of volcanic sulfur input.

**Linkages between massive volcanic events and the mid-Cretaceous oceanic anoxic events.** Previous studies have revealed that the onsets of the major Cretaceous OAEs (OAE1a, Wezel, Fallot, and OAE2) in the Tethyan region correspond to unradiogenic Os isotopic shifts[13,16,19–21], which is compatible with synchronicity between massive submarine volcanism and OAEs. During these OAEs, unradiogenic Os shifts are often accompanied by the negative carbon isotopic excursions[19–21,29], implying the volcanic events supply mantle-derived $CO_2$ with negative carbon isotopic values. Besides, a 2–16 times increase in the input of mantle-derived Os is required to explain these unradiogenic Os isotopic shifts. Considering that Os could have been supplied in highly volatile oxidized form ($OsO_4$), enormous amounts of other volatile trace metal elements could have been also injected into the ocean-atmosphere system during the most prominent unradiogenic Os isotopic shifts in these OAEs (OAE1a, Wezel and Fallot events, and OAE2). This possibility supports the linkage between bio-limiting trace metal input and the high productivity[53]. The proposed triggering mechanism of the OAEs is as follows[17]: massive volcanic events released large quantities of greenhouse gases into the atmosphere that caused an increase in the temperature; as a result, enhanced continental weathering supplied the nutrients to the ocean that led to enhanced primary productivity and ultimately to ocean eutrophication[17]. In addition, the volcanic activity could have provided iron and other bio-limiting trace metals to the ocean, which might have further stimulated the primary productivity[17,53]. In addition, the warming of deep-, intermediate-, and surface- water could have disrupted the thermocline, which triggered sustained upwelling and maintained the high primary productivity[17]. The decomposition of a large amount of organic matter at the seafloor consumed oxygen and expanded the oxygen minimum zones.

Our Os isotopic data revealed that multiple volcanic signals also correspond to the base and top of OAE1c (Figs. 2 and 3a); however, the unradiogenic shift does not cover the most prominent organic-rich interval of the OAE1c, called the Amadeus Segments (Figs. 2 and 3a). In addition, $^{187}Os/^{188}Os$ values do not show any significant fluctuation during OAE1d, which suggests the absence of intensive submarine volcanic activity. Therefore, we consider that the onsets of OAE1c and OAE1d were unrelated to massive submarine volcanism, unlike other major mid-Cretaceous OAEs. A mercury anomaly has been reported just below the OAE1d horizon at the Youxia section, the eastern Tethys, which has been interpreted as the submarine

volcanic eruption at the Kerguelen Plateau[54]. However, considering the lack of Os isotopic variations around OAE1d, this mercury enrichment is probably more related to local perturbations with limited influence on global climate. Major Cretaceous OAEs (OAE1a, Wezel, Fallot, and OAE2) are represented by thick (~6 cm to 2 m) organic-rich intervals, whereas the sedimentary expression of OAE1c and OAE1d in the Umbria–Marche Basin consist of cyclic alternations of thin black shales[8]. Similar cyclic intercalations of thin black shale layers in a carbonate sequence have been observed in the Valanginian–Barremian, Albian, and upper Cenomanian in the Umbria–Marche Basin (Fig. 3)[8,22]. During the Quaternary, astronomically modulated monsoonal activity cyclically enhanced the hydrology of the Mediterranean Sea at low latitude, which supplied freshwater and nutrients to the peri-continental ocean[55]. The resulting input of terrigenous organic matter, stratification, and slightly enhanced productivity led to oxygen-depleted bottom-water conditions and the deposition of organic-rich sediments dominated by terrigenous sources[56]. Thus, the lack of the unradiogenic Os isotopic shift and the cyclic deposition of thin black shale layers during OAE1c and OAE1d may suggest a regional-scale weak marine anoxia caused by monsoonal activity modulated by astronomical cycles as proposed by previous studies[56] rather than an episodic large volcanic event[54]. The increase in primary productivity was not significant in the Tethyan region during OAE1d[56]. However, a small positive carbon isotopic excursion during OAE1d suggests a slight increase in the primary production (Fig. 2). In addition, organic-rich sediments are reported from the Calera Limestone in California, which was deposited in the Pacific Ocean, and thus the oxygen-depleted condition could have prevailed in the East Pacific as well[14]. Thus, the latter process can also cause a supra-regional increase in productivity to some extent.

The organic geochemical properties of mid-Cretaceous black shale horizons are also consistent with differences in the origins of mid-Cretaceous OAEs. Erbacher et al.[57] proposed to classify mid-Cretaceous OAEs into two types on the basis of organic geochemistry and radiolarian occurrences: (1) productivity (P-) OAEs (e.g., OAE1a, OAE1d, and OAE2); and (2) detrital (D-) OAEs (e.g., OAE1c). Organic matter deposited in the Umbria–Marche Basin during OAE1a and OAE2 is close to Type II kerogen, which is derived from marine organisms[57], whereas the organic matter of the OAE1c and OAE1d black shales is classified as Type III kerogen, which has a continental origin[57]. However, OAE1d and part of OAE2 are classified as P-OAE, although their organic matter was identified as Type III kerogen of continental origin[57]. To solve this contradiction, the classification of OAE types should be modified. Thus, on the basis of the organic geochemistry and Os isotopic data, we here propose to classify the mid-Cretaceous OAEs into: (1) volcanic-induced OAEs triggered by episodic burial of organic-rich sediments derived from marine organisms; and (2) monsoon-induced OAEs that are mainly caused by water-mass stratification triggered by freshwater input caused by the cyclic intensification of monsoonal activity. The organic matter of monsoon-induced OAEs is mainly composed of terrestrial materials and an increase in productivity is less significant[56].

Among the mid-Cretaceous OAEs, OAE1b is a problematic example. In the Umbria–Marche Basin OAE1b is composed of several major organic-rich horizons (Jacob, Kilian, Urbino, and Leenhardt Levels) intercalated with numerous thin black shale horizons[12]. Although the short unradiogenic Os isotopic shifts have been reported around the Kilian Level, other black shale horizons lack unradiogenic Os isotopic shifts[16]. Besides, the upper part of the OAE1b is characterized by the temporal radiogenic Os isotopic excursions, which constitute a different feature from other mid-Cretaceous OAEs. Since OAE1b

continued for several million years and contains different types of organic-rich sediments, we considered that OAE1b may be a mixture of volcanic- and monsoon-induced OAEs.

**Cause of the temperature variations during the mid-Cretaceous.** The mid-Cretaceous has often been regarded as a warm geological interval caused by high $pCO_2$, which was sustained by enhanced hydrothermal activity associated with oceanic crustal production[7]. Considering the extremely high temperature and $pCO_2$ during major OAEs (OAE1a and OAE2)[3,58,59], this model seems correct over a short time scale. However, the model cannot explain the long-term temperature variations of the mid-Cretaceous. The $^{187}Os/^{188}Os$, $^{87}Sr/^{86}Sr$, and $\delta^{34}S$ records suggest an intensification of hydrothermal activity during the Aptian, corresponding to the relatively cool interval during the mid-Cretaceous[3,42] (Fig. 3). Furthermore, the highest temperature during the mid-Cretaceous was recorded during the Cenomanian–Turonian. However, no long-term hydrothermal activity associated with LIPs formation has been reported during the Cenomanian (Fig. 3).

This contradiction may be explained by the location and style of the volcanic activity. When a basaltic plateau was emplaced under submarine conditions, outgassing from submarine volcanism and the expansion of the volatile to shallower waters could have been suppressed by high hydrostatic pressure[60], and, thus, they may not have contributed to the long-term increase in the $pCO_2$. Indeed, most of OJN was emplaced under submarine conditions during the Aptian and may not have caused a long-term increase in $pCO_2$. The temperature started to increase at the Aptian–Albian boundary and reached a maximum at the Cenomanian, which corresponded to the subaerial eruption of the Kerguelen Plateau (Fig. 3). As most of the Kerguelen Plateau was emplaced under subaerial conditions, a large amount of $CO_2$ could have been directly released into the atmosphere, and could have contributed to the increase in $pCO_2$ and temperature. During the Cenomanian, subaerial volcanic eruptions occurred at Kerguelen[39–41] and the High Arctic Large Igneous Province[61], which could have caused the increase in temperature during the Cenomanian–Turonian interval. In addition, subaerial volcanic activity in the circum-Pacific region was active during the mid-Cretaceous. For example, the volcanic events associated with the formation of Japanese granitoids were most active during the Cenomanian to Turonian (95–80 Ma) (Fig. 3f)[62], approximately corresponding to the mid-Cretaceous thermal maximum. At the Cenomanian–Turonian boundary, the massive submarine volcanic eruptions at the Caribbean Plateau could have further contributed to the intensification of global warming during the OAE2[63]. We conclude that massive and continuous subaerial volcanism could have contributed to the high $pCO_2$ and high temperature, whereas submarine volcanism had a minor effect on the long-term increase in temperature. The low $pCO_2$ interval during the late Aptian is associated with the substantial carbonate phytoplankton production called *Nannoconus truitti* acme[53]. Therefore, the eruption style (e.g., submarine or subaerial) and its duration could have potentially influenced not only the temperature variations but also the diversity of calcareous planktons during the mid-Cretaceous.

In conclusion, our $^{187}Os/^{188}Os_i$ values showed the pronounced unradiogenic shifts during the early to mid-Aptian, late Albian, and end-Cenomanian, reflecting intensive hydrothermal activity associated with the formation of LIPs. In addition, temporal radiogenic $^{187}Os/^{188}Os_i$ shifts observed during the early Albian can be interpreted as an enhancement of continental weathering. During the mid-Cretaceous, the OAEs are classified into: (1) volcanic-induced OAEs (e.g., OAE1a, Wezel Level, Fallot Level,

and OAE2); and (2) monsoon-induced OAEs (OAE1c and OAE1d). The warmest interval during the mid-Cretaceous corresponded to a phase of enhanced subaerial volcanic episodes with no evidence of long-term intensive hydrothermal activity. Thus, we conclude that subaerial volcanic episodes and the following outgassing were the main cause of the warm mid-Cretaceous climate.

## Methods

**Re-Os analysis.** We followed the analytical methods described by Matsumoto et al.[16]. Cleaned sedimentary rocks were powdered in an agate mill. After spiking with $^{190}Os$- and $^{185}Re$-rich solutions, Re and Os of the carbonate rocks were extracted by the inverse aqua regia (mixture of 30 wt% HCl 1 ml and 68% HNO$_3$ 3 ml) under 240 °C for 48 h. After Os was purified by CCl$_4$ extraction, HBr extraction, and micro-distillation, Os abundances and isotopic compositions were determined by negative thermal ionization–mass spectrometry (TRITON, Thermo Fisher Scientific, USA) at the Japan Agency for Marine-Earth Science and Technology (JAMSTEC, Japan). The abundances of Re were determined by a quadrupole inductively coupled plasma-mass spectrometer (iCAP Qc, Thermo Fisher Scientific, USA) at JAMSTEC. Initial $^{187}Os/^{188}Os$ values ($^{187}Os/^{188}Os_i$) were calculated from the measured $^{187}Os/^{188}Os$ and $^{187}Re/^{188}Os$ values, the estimated ages (Supplementary Tables 1–3), and the $^{187}Re$ decay constant ($1.666 \times 10^{-11}$ yr$^{-1}$[64]). The average procedural blanks of Os and $^{187}Os/^{188}Os$ were $0.8 \pm 0.5$ pg and $0.13 \pm 0.04$, respectively. The average Re-procedural blank was $14 \pm 11$ pg.

**Stable carbon isotopic ratio of carbonate.** The stable carbon isotope ratio of carbonate ($\delta^{13}C_{carb}$) was measured with an isotope ratio-mass spectrometer (Delta V plus, Thermo Fisher Scientific, USA), equipped with an automated carbonate reaction device (GasBench II, Thermo Fisher Scientific, USA), at the Atmosphere and Ocean Research Institute, University of Tokyo (Japan). Isotopic values are reported in delta notation with respect to PeeDee Belemnite (PDB), based on an NBS-19 value of $+1.95$‰ for $\delta^{13}C$. External reproducibility was estimated from repeated analysis of the NBS-19 standard ($n = 20$) within an analytical batch; the typical values were better than 0.05‰ and 0.08% for $\delta^{18}O$ and $\delta^{13}C$, respectively (1 SD) (Supplementary Tables 2 and 3).

## Data availability

The authors declare that the Os and carbon isotopic data generated in this study are provided in the Supplementary Information.

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

## Acknowledgements

We thank Dr. K. Suzuki, Dr. T. Nozaki, and Y. Otsuki for their support in Re-Os analysis. We express sincere gratitude to K. Tanaka and N. Izumoto for their support in $\delta^{13}C_{carb}$ analysis. We warmly thank Dr. K. Matsuzaki for the constructive advice on the manuscript. This study was financially supported by Grant-in-aid for JSPS Research Fellow (19J20708) and JSPS KAKENHI (21H01203). R.C. and R.T. acknowledge the FUSP (Fundação de Apoio à Universidade de São Paulo)-Petrobras BARREMAG and 2405 projects for the financial support of the PLG core drilling.

## Author contributions

H.M. conceived and designed this work and wrote the original manuscript. H.M., R.C., F.F., L.J., R.T. and J.F.S. collected samples. H.M. and J.K. conducted Re-Os analysis. H.N. and K.S. conducted the carbon isotopic analysis. All authors discussed and interpreted the results and contributed to the manuscript.

## Competing interests

The authors declare no competing interests.
