## [Peer Review File · Nature Communications]

Mid-Cretaceous marine Os isotope stratigraphy for the evolutionary history of hydrothermal activityREVIEWER COMMENTS:

Reviewer #1 (Remarks to the Author):

Dear authors,

The manuscript is well written. The results are well summarized. The paper is well structured. Nice story!

There are only a few points that I do not find so successful. The interpretations for the different intervals are not always consistent. For example, I would like you to look at the OAE 1b in a more differentiated way. I think there is more music in your data than you express. The statement that OAE 1b is to be regarded like OAE 2 or the Aptian OAEs is too simple. Jacob seems to be below the negative excursion. Kilian is characterized by a clear volcanogenic signal. For Leenhardt and Co the data is weak. I propose to discuss this time interval in more detail .

Your statement that submarine volcanism does not contribute to the CO₂ rise in the atmosphere is brave! Is there any evidence for this - perhaps references from recent submarine volcanism or modeling data? One should not let this statement stand alone and undiscussed.

I noted a few less important things in the manuscript.

Reviewer #2 (Remarks to the Author):

The manuscript by Matsumoto and co-authors provides the first compilation of Os₁₈₇/Os₁₈₈(i) record across the Late Barremian-Late Cenomanian time interval by combining many published data and new data collected for the late Albian-early Cenomanian.

The dataset is very good and represents an important reference for this time interval. The work has potential for publication. Please find below my general comments and some minor comments/suggestions in the pdf attached.

General comment

The main issue I see is that the authors can improve the discussion with respect to what we already know from other works about the paleoceanographic conditions occurred during this time interval. The discussion chapter lacks a deeper discussion with respect to literature especially regarding the ocean-atmosphere system evolution across the mid-Cretaceous and the studied OAEs.

lines 248-257: this is not something new. Moreover, literature data do not support a "productivity" model to explain all the OAEs mentioned by the authors. I suggest to revise this part taking into account what is known from literature and, if possible, also trying to go a bit further with the interpretation since now the authors have a nice long-term curve of the Os ratio variations.

lines 264-274: The new datasets confirms what has already been said in some published works about "OAE 1 c" and OAE 1d which were not induced by volcanic activity but rather by other imposed paleoclimatic and paleoecological conditions. I suggest to revise this part underlying this aspect and citing the papers who already said that on the basis of other proxies. Please, be also careful that, following some works, OAE 1d black shales were not marked by high productivity (see my comments in the text)

Figure 3

Which time scale did you use? This is critical with respect to the position of the LIP ages. Please specify and eventually discuss that in the text.

Reviewer #3 (Remarks to the Author):

The authors have recently published Os-isotope data from the Cretaceous of Marche Umbria (Poggio le Guaine core) covering much of the Aptian–Albian interval, and there are pre-existing data from various parts of the world for the OAE 1a and OAE 2 intervals, both of which show major shifts to relatively unradiogenic values of Os and Sr isotopes (e.g. Turgeon and Creaser, Du Vivier et al., Jones and Jenkyns, Tejada et al., Bottini et al.: well reviewed in Dickson et al in *Large Igneous Provinces: A Driver of Global Environmental and Biotic Changes*, Geophysical Monograph 255). The authors put all this together with some new data from the Cenomanian of the Bottaccione Gorge near Gubbio and an ODP Site on the Exmouth Plateau off Australia to create some relatively long chemostratigraphic time series, in which they can identify a number of Os-isotope excursions of likely global significance. The authors discuss the putative roles of volcanism and hydrothermal activity in governing the Os-isotope signatures of seawater and endorse the currently accepted interpretations in the literature. Because most people have focused on ‘events’ in their search for geochemical anomalies, the background has tended to be ignored (and may yet produce some surprises), so more extensive time series are to be welcomed. However, I’m not sure whether the simple two-fold division of OAEs into volcanic- and monsoon-related holds up to scrutiny (see below). Further, the authors have ignored some key literature that goes against their argumentation, thereby weakening the impact of the paper.

Line 44: ‘palaeo-’ or ‘paleo-’? Be consistent! (‘palaeo-’) in British English

Line 45: TEX86 data from the uppermost Cretaceous of the Arctic Ocean suggest sea-surface temperatures in excess of 15°C.

Line 57: Robinson et al. recognize an OAE1d black shale in the Calera Limestone of California, which derives from the paleo-Pacific (*Geological Society of America Bulletin*, 120, pp.1416-1426). Hence, the reach of this event may be greater than just Tethys/Atlantic. The Pacific also has a record of OAE 1a and OAE 2 black shales that reflect major events. How does this Calera Limestone occurrence affect the interpretation of OAE 1d as monsoon-related? Note also the new mercury evidence bearing on OAE 1d: Yao, H., Chen, X., Yin, R., Grasby, S.E., Weissert, H., Gu, X. and Wang, C., 2021. Mercury evidence of intense volcanism preceded oceanic anoxic event 1d. *Geophysical Research Letters*, 48, p.e2020GL091508. Some reconsideration is warranted here, even if the absence of an Os-isotope signal may be telling us something.

Line 104: I agree that considering OAE 1b as a cluster of events is the most useful approach. The Kilian is now taken to mark the Aptian-Albian boundary (GSSP: see Kennedy et al., *Episodes*, vol. 40).

Line 107: it is not clear to me what ‘partially records OAE 1c’ means. How is (or was) OAE 1c defined? Maybe go back to Arthur, M.A., et al., 1990. *Stratigraphy, geochemistry, and paleoceanography of organic carbon-rich Cretaceous sequences. In Cretaceous resources, events and rhythms* (pp. 75-119). – who suggested the OAE nomenclature.

Line 132: Reference 9 does not discuss carbon-isotope data. Refer to the paper of Gambacorta et al. (*Newsletters on Stratigraphy*, 48, 299-323) for carbon-isotope patterns in the Upper Cretaceous of Marche–Umbria.

Line 184: I could not find a reference to modelling of weathering of Ontong Java in Tejada et al.,

although they suggest various types of basalt–seawater interaction to produce the Os-isotope excursion . Presumably warm- to low-temperature submarine weathering of basalt could be part of this.

Line 189: during the early Albian

Line 192: belemnites (plural)

Line 199:supports

Line 217: in what way is the geochemical behaviour of Os different from that of Sr in this context?

Line 228: presumably you exclude the possibility that the positive sulphur-isotope excursion was due to increased pyrite burial in black shales?? As illustrated in Fig. 3c, most of the increase in S-isotope values took place in the Cenomanian, post-dating OAE 1d; in so doing, S isotopes rise in parallel with established carbon-isotope curves (e.g. Jarvis et al. for the English Chalk) likely signifying gradually increasing global marine carbon burial in the run-up to OAE 2. The alternative to the idea of a decrease in hydrothermal input of sulfur, namely that the data reflect synchronous increased burial of isotopically light carbon and sulphur, needs to be discussed.

Line 244: Sulfur-isotope evolution around OAE 2 (data from the English Chalk and the Scaglia in Marche–Umbria) is discussed in detail in papers by Jeremy Owens et al. (PNAS, 2013 and Sedimentology, 2017) and Jenkyns et al. (Sedimentology, 2017): the pattern is complex, with relatively high values post-OAE 2 recorded in both pyrite and CAS – exactly the opposite of what is described in the text by the authors. The Paytan et al. reference given is inappropriate.

Line 248: Most people would not consider the Wezel and Fallot Events as major OAEs. How extensive is their sedimentary record outside Europe?

Line 249: 'is compatible with' rather than 'suggests'.
Association does not prove cause and effect.

Lines 249–256: This paragraph is presented as if it is novel – but the ideas presented have been in the literature for many years.

Lines 262–263 see comments above as to evidence for volcanic activity around OAE 1d. See also: Wilson, P.A. and Norris, R.D., 2001. Warm tropical ocean surface and global anoxia during the mid-Cretaceous period. *Nature*, 412(6845), pp.425-429 – who provide documentary evidence for the global reach of OAE 1d, extending into the Pacific.

Lines 316–317: what about the Caribbean Plateau?

Line 374: change 'The since' to 'Sincere'

References: many abbreviations are not complete and/or not compatible with journal style. Special Publications/Papers (e.g. Fassell and Bralower; Savian et al.) are not clearly distinguished. Other examples:

Line 526: reference should begin: Pogge von Strandmann, P.A.E. (see also line 488)

Line 546: reference incomplete

REVIEWER COMMENTS

Reviewer #2 (Remarks to the Author):

The authors have provided an improved version of their work which takes into account the suggestions of the reviewers. I have especially found improvements in the discussion which are now more complete. I do recommend the publication.

In the current version I spotted a few minor things reported as follows:

lines 165 and 167 "represent" is repeated twice.

line 240 "is the more" I do not think this is correct in english, please check

line 270 "Previous studies have revealed THAT the onsets of the major Cretaceous OAEs (OAE1a, Wezel, Fallot, and OAE2) in the Tethyan region correspond to...."

Fig. 3 "Furlo" is misspelled and box "e" and "f" are not aligned

Reviewer #3 (Remarks to the Author):

The authors have addressed the most significant points raised in the reviews. I note some minor points in the text that need attention and some more substantive issues.

Line 242: change 'cease' to 'cessation'

Line 273: change to 'volcanism and OAEs.'

Line 281: change 'mental' to 'metal'

Line 299: change to 'A mercury anomaly . . .'

Line 309: change to 'During the Quaternary . . .'

Line 381: look at: Scaife, J.D. et al., 2017. Sedimentary mercury enrichments as a marker for submarine large igneous province volcanism? Evidence from the Mid-Cenomanian event and Oceanic Anoxic Event 2 (Late Cretaceous). *Geochemistry, Geophysics, Geosystems*, 18, 4253–4275.

The mercury evidence is instructive in terms of proposed submarine/subaerial LIPs.

Page 21: in their discussion of Cenomanian–Turonian temperatures the authors state that no long-term evidence for hydrothermal activity is recorded and they reference Fig. 3. However, there is very little in the way of an illustrated Turonian record in this figure and notably marine Sr-isotope keep falling from the OAE 2 interval (big shift) to less radiogenic values over the whole of the Turonian (see compilation in Jones and Jenkyns (*American Journal of Science*) and papers by McArthur et al. Maybe the Caribbean, High Arctic and other LIPs were supplying non-radiogenic Sr (and Os?) over this period?

Figure 3: the two right-hand columns are offset from the other three. Should they not line up?

Reviewer #1

General comments

1-1. Comment

The manuscript is well written. The results are well summarized. The paper is well structured. Nice story!

There are only a few points that I do not find so successful. The interpretations for the different intervals are not always consistent. For example, I would like you to look at the OAE 1b in a more differentiated way. I think there is more music in your data than you express. The statement that OAE 1b is to be regarded like OAE 2 or the Aptian OAEs is too simple. Jacob seems to be below the negative excursion. Kilian is characterized by a clear volcanogenic signal. For Leenhardt and Co the data is weak. I propose to discuss this time interval in more detail.

Reply

Thank you for many valuable comments. As you pointed out, OAE1b is composed of many organic-rich layers having different geochemical features. Thus, we have dealt with OAE1b separately in the different paragraphs and added a more detailed explanation as following “*Among the mid-Cretaceous OAEs, OAE1b is a problematic example. In the Umbria–Marche Basin OAE1b is composed of several major organic-rich horizons (Jacob, Kilian, Urbino, and Leenhardt Levels) intercalated with numerous thin black shale horizons¹². Although the short unradiogenic Os isotopic shifts have been reported around the Kilian Level, other black shale horizons lack unradiogenic Os isotopic shifts¹⁶. Besides, the upper part of the OAE1b is characterized by the temporal radiogenic Os isotopic excursions, which is a different feature from other mid-Cretaceous OAEs. Since OAE1b continued for several million years and contains different types of organic-rich sediments, we considered that OAE1b may be a mixture of volcanic- and monsoon-induced OAEs.*” (L. 342–350).

1-2. Comment

Your statement that submarine volcanism does not contribute to the CO₂ rise in the atmosphere is brave! Is there any evidence for this - perhaps references from recent submarine volcanism or modeling data? One should not let this statement stand alone and undiscussed.

Reply

Thank you for the comments. The volcanic events under the deep-sea condition were totally different from the subaerial eruption and the outgassing of volatile is suppressed

by the hydrostatic pressures. We added the recent articles explaining the difference between the subaerial and submarine volcanism and added some explanations on this point as “*When a basaltic plateau was emplaced under submarine conditions, outgassing from submarine volcanism and the expansion of the volatile to shallower waters could have been suppressed by high hydrostatic pressure⁵⁹, and, thus, they may not have contributed to the long-term increase in the $p\text{CO}_2$.*” (L. 363–366).

Minor comments

1-3. Comment

L.1: “This title is misleading a bit. Why not mentioning the first long term Os-Isotope record”

Reply

Thank you for the suggestion. We modified the title of this manuscript as “*Mid-Cretaceous marine Os isotope stratigraphy: evolutionary history of hydrothermal activity*” (L. 1)

1-4. Comment

L. 140-142: “I am not sure this makes a lot of sense. Of course you may have a high percentage of CaCO_3 based solely on the presence of authigenic carbonates. Better delete as the sentence doesn't contribute to your story anyway”

Reply

L.146. Thank you for the suggestion. We removed this part.

1-5. Comment

L. 248: “Looks as if the Os-isotopes decrease happens after Niveau Jacob, right? I've never been a big fan of grouping the different horizons together one OAE 1b (and then differentiating between OAE 1a and Livello Wezel). Maybe you should try to tell a more differentiated story for the different black shale levels based on your data - now. The evidence of negative Os isotope values is clear for Killian but to my opinion not clear at all for the other black shale levels during OAE 1b...”

Reply

As you have suggested, OAE1b is composed of several black shale horizons having different geochemical features and possibly different origins. Thus, organic-rich sediments should be separately discussed. Thus, we added a more detailed discussion of OAE1b in the different sections as “*Among the mid-Cretaceous OAEs, OAE1b is a problematic example. In the Umbria–Marche Basin OAE1b is composed of several major*

organic-rich horizons (Jacob, Kilian, Urbino, and Leenhardt Levels) intercalated with numerous thin black shale horizons¹². Although the short unradiogenic Os isotopic shifts have been reported around the Kilian Level, other black shale horizons lack unradiogenic Os isotopic shifts¹⁶. Besides, the upper part of the OAE1b is characterized by the temporal radiogenic Os isotopic excursions, which is a different feature from other mid-Cretaceous OAEs. Since OAE1b continued for several million years and contains different types of organic-rich sediments, we considered that OAE1b may be a mixture of volcanic- and monsoon-induced OAEs.” (L. 342–350).

1-5. Comment

L. 265: “see my comment in Line 248”

Reply

As we explained above, we have discussed OAE1b in the different sections (L. 342–350).

1-6 Comment

L. 307-311: “I am not an expert of submarine volcanism and its influence on pCO₂, but here I would like to see some quotes that confirm this theory!”

Reply

L. 317. Thank you for the comment. We added a detailed explanation and a reference of a recent article explaining the difference between the submarine and subaerial eruptions supporting our hypothesis as “*When a basaltic plateau was emplaced under submarine conditions, outgassing from submarine volcanism and the expansion of the volatile to shallower waters could have been suppressed by high hydrostatic pressure⁵⁹, and, thus, they may not have contributed to the long-term increase in the pCO₂.*” (L. 363–366).

Reviewer #2

The manuscript by Matsumoto and co-authors provides the first compilation of Os¹⁸⁷/Os¹⁸⁸(i) record across the Late Barremian-Late Cenomanian time interval by combining many published data and new data collected for the late Albian-early Cenomanian.

The dataset is very good and represents an important reference for this time interval. The work has potential for publication. Please find below my general comments and some minor comments/suggestions in the pdf attached.

General comment

2-1. Comment

The main issue I see is that the authors can improve the discussion with respect to what we already know from other works about the paleoceanographic conditions occurred during this time interval. The discussion chapter lacks a deeper discussion with respect to literature especially regarding the ocean-atmosphere system evolution across the mid-Cretaceous and the studied OAEs.

Reply

Thank you for the comment. We added discussions on the detailed atmosphere and oceanographic conditions during OAEs and mid-Cretaceous and added some references as:

(L. 270-292) *“Previous studies have revealed the onsets of the major Cretaceous OAEs (OAE1a, Wezel, Fallot, and OAE2) in the Tethyan region correspond to unradiogenic Os isotopic shifts^{13,16,19-21}, which is compatible with synchronicity between massive submarine volcanism and OAE. During these OAEs, unradiogenic Os shifts are often accompanied by the negative carbon isotopic excursions^{19-21, 29}, implying the volcanic events supply mantle-derived CO₂ with negative carbon isotopic values. Besides, 2 to 16 times increase in the input of mantle-derived Os are required to explain these unradiogenic Os isotopic shifts. Considering that Os could have been supplied in highly volatile oxidized form (OsO₄), enormous amounts of other volatile trace metal elements could have been also injected into the ocean-atmosphere system during the most prominent unradiogenic Os isotopic shifts in these OAEs (OAE1a, Wezel and Fallot events, and OAE2). This possibility supports the linkage between bio-limiting trace metal input and the high productivity⁵². The proposed triggering mechanism of the OAEs is as follows¹⁷: massive volcanic events released large quantities of greenhouse gases into the atmosphere that caused an increase in the temperature; as a result, enhanced continental weathering supplied the nutrients to the ocean that led to enhanced primary productivity and ultimately to ocean eutrophication¹⁷. In addition, the volcanic activity could have provided iron and other bio-limiting trace metals to the ocean, which might have further stimulated the primary productivity^{17,52}. In addition, the warming of deep-, inter-mediate, and -surface ocean could have disrupted the thermocline, which triggered sustained upwelling and maintained the high primary productivity¹⁷. The decomposition of a large amount of organic matter at the seafloor consumed oxygen and thickened the oxygen minimum zones.”* and

(L. 382-386) *“The low pCO₂ interval during the late Aptian is associated with the substantial carbonate phytoplankton production called Nannoconus truitti acme⁵². Therefore, the eruption style (e.g., submarine or subaerial) and its duration could have potentially influenced not only the temperature variations but also the diversity of*

calcareous planktons during the mid-Cretaceous.”.

2-2. Comment

lines 248-257: this is not something new. Moreover, literature data do not support a “productivity” model to explain all the OAEs mentioned by the authors. I suggest to revise this part taking into account what is known from literature and, if possible, also trying to go a bit further with the interpretation since now the authors have a nice long-term curve of the Os ratio variations.

Reply

As you have indicated, the references do not fully support that these OAEs are productivity OAEs. Thus, we modified the meaning of this sentence to avoid misleading interpretation as following (L. 270–273) “*Previous studies have revealed the onsets of the major Cretaceous OAEs (OAE1a, Wezel, Fallot, and OAE2) in the Tethyan region correspond to unradiogenic Os isotopic shifts^{13,16,19–21}, which is consistent with synchronicity between massive submarine volcanism and OAE.”.*

In addition, we added the more precise discussion on the Os behavior and its implication of the emission of the volatile elements and its relation to onset of OAEs as “*During these OAEs, unradiogenic Os shifts are often accompanied by the negative carbon isotopic excursions^{19–21, 29}, implying the volcanic events supply mantle-derived CO₂ with negative carbon isotopic values. Besides, 2 to 16 times increase in the input of mantle-derived Os are required to explain these unradiogenic Os isotopic shifts. Considering that Os could have been supplied in highly volatile oxidized form (OsO₄), enormous amounts of other volatile trace metal elements could have been also injected into the ocean-atmosphere system during the most prominent unradiogenic Os isotopic shifts in these OAEs (OAE1a, Wezel and Fallot events, and OAE2). This possibility supports the linkage between bio-limiting trace metal input and the high productivity⁵².” (L. 273-282).*

2-3. Comment

lines 264-274: The new datasets confirms what has already been said in some published works about “OAE 1 c” and OAE 1d which were not induced by volcanic activity but rather by other imposed paleoclimatic and paleoecological conditions. I suggest to revise this part underlying this aspect and citing the papers who already said that on the basis of other proxies. Please, be also careful that, following some works, OAE 1d black shales were not marked by high productivity (see my comments in the text)

Reply

Thank you for the comment. We added the explanation on the previous studies and

mentioned the productivity during OAE1d as following (L. 295-321) “Therefore, we consider that the onsets of OAE1c and OAE1d were unrelated to massive submarine volcanism, unlike other major mid-Cretaceous OAEs. Mercury anomaly has been reported just below the OAE1d horizon at the Youxia section, the eastern Tethys, which has been interpreted as the submarine volcanic eruption at the Kerguelen Plateau⁵³. However, considering the lack of Os isotopic perturbations around OAE1d, this mercury enrichment is probably more related to local perturbation with limited influence on global climate. Major Cretaceous OAEs (OAE1a, Wezel, Fallot, and OAE2) are represented by thick organic-rich intervals, whereas the sedimentary expression of OAE1c and OAE1d in the Umbria–Marche Basin consist of cyclic alternations of thin black shales⁸. Similar cyclic intercalations of thin black shale layers in a carbonate sequence have been observed in the Valanginian–Barremian, Albian, and upper Cenomanian in the Umbria–Marche Basin (Fig. 3)^{8,22}. During Quaternary, astronomically modulated monsoonal activity cyclically enhanced the hydrology of the Mediterranean Sea at low latitude, which supplied freshwater and nutrients to the peri-continental ocean. The resulting input of terrigenous organic matter, stratification, and slightly enhanced productivity led to oxygen-depleted bottom-water conditions and the deposition of organic-rich sediments dominated by the terrigenous origin^{54–55}. Thus, the lack of the unradiogenic Os isotopic shift and the cyclic deposition of thin black shale layers during OAE1c and OAE1d may suggest a regional-scale weak marine anoxia caused by monsoonal activity modulated by astronomical cycles as proposed by previous studies⁵⁵ rather than an episodic large volcanic event⁵³. The increase in the primary productivity was not significant at the Tethyan region during OAE1d⁵⁵. However, a small positive carbon isotopic excursion during OAE1d suggests a slight increase in the primary production (Fig. 2). In addition, organic-rich sediments are reported from the Calera Limestone in California, which was deposited in the Pacific Ocean, and thus the oxygen-depleted condition could have prevailed in the East Pacific as well¹⁴. Thus, the latter process can also cause a supra-regional increase in productivity to some extent.”.

2-4. Comment

Figure 3

Which time scale did you use? This is critical with respect to the position of the LIP ages. Please specify and eventually discuss that in the text.

Reply

Figure. 3: We used the time scale of Gale et al. (2020) and added the reference on this point (L. 692) and include it in the caption.

Minor comments

2-5. Comment

L. 25: “capital letters”

Reply

L. 26. We modified “oceanic anoxic events” to “Oceanic Anoxic Events”.

2-6. Comment

L. 65: “I sugget to add Erba 2004”

Reply

L. 67: We added the reference in this part.

2-7. Comment

L. 168: “On the contrary,”

Reply

L. 171–172: We changed “However” to “On the contrary”.

2-8. Comment

L. 174: “you may add also Bauer et al. 2021 published on Geology "Pulsed volcanism and rapid oceanic deoxygenation during Oceanic Anoxic Event 1a. ""

Reply

L.179. Thank you for introducing us the recent paper. We have cited it.

2-9. Comment

L. 222: “may be you meant "when"? this is not clear, please check this sentence”

Reply

L.228. Thank you for the comment. We changed “where” to “when”.

2-10. Comment

L. 253-254: “is it this really applicable to all OAEs mentioned above? Some literature data do not agree with this interpretation and/or is much more complicated than that. I suggest that the authors revise this part of the discussion by taking into account what we know from literature and provide a more thoughtful interprtation”

Reply

Thank you for the comments we added more explanation and discussions on the mechanism of volcanic-induced OAEs as (L. 282–292) “*The proposed triggering*

mechanism of the OAEs is as follows¹⁷: massive volcanic events released large quantities of greenhouse gases into the atmosphere that caused an increase in the temperature; as a result, enhanced continental weathering supplied the nutrients to the ocean that led to enhanced primary productivity and ultimately to ocean eutrophication¹⁷. In addition, the volcanic activity could have provided iron and other bio-limiting trace metals to the ocean, which might have further stimulated the primary productivity^{17,52}. In addition, the warming of deep-, inter-mediate, and -surface ocean could have disrupted the thermocline, which triggered sustained upwelling and maintained the high primary productivity¹⁷. The decomposition of a large amount of organic matter at the seafloor consumed oxygen and thickened the oxygen minimum zones.” Besides, we discussed the OAE1b separately in the different paragraph because OAE1b contains many different types of black shale horizons and origins: (L. 342–350) “Among the mid-Cretaceous OAEs, OAE1b is a problematic example. In the Umbria–Marche Basin OAE1b is composed of several major organic-rich horizons (Jacob, Kilian, Urbino, and Leenhardt Levels) intercalated with numerous thin black shale horizons¹². Although the short unradiogenic Os isotopic shifts have been reported around the Kilian Level, other black shale horizons lack unradiogenic Os isotopic shifts¹⁶. Besides, the upper part of the OAE1b is characterized by the temporal radiogenic Os isotopic excursions, which is a different feature from other mid-Cretaceous OAEs. Since OAE1b continued for several million years and contains different types of organic-rich sediments, we considered that OAE1b may be a mixture of volcanic- and monsoon-induced OAEs.”.

2-11. Comment

L. 255: “I suggest to add Erba et al 2015.”

Reply

L. 288 We added the citation here.

2-12. Comment

L. 269: at →in”

Reply

L. 309. We changed “at” to “in”.

2-13. Comment

L. 272-273: “be careful as Bornemann et al 2005 and Gambacorta et al 2020, for example, do not find high productivity during OAE 1d.”

Reply

Thank you for the comments. As you have pointed out, there are few data suggesting the high productivity during OAE1d and 1c. However, in case of OAE1d, the possibility of slightly enhanced productivity cannot be ruled out because it is accompanied by the slight positive carbon isotopic excursion and organic-rich sediments have been discovered in the Pacific region. Therefore, we modified “high productivity” to “slightly enhanced productivity” (L. 313) and added more detailed explanation on this point as (L. 298-324) “*Therefore, we consider that the onsets of OAE1c and OAE1d were unrelated to massive submarine volcanism, unlike other major mid-Cretaceous OAEs. Mercury anomaly has been reported just below the OAE1d horizon at the Youxia section, the eastern Tethys, which has been interpreted as the submarine volcanic eruption at the Kerguelen Plateau⁵³. However, considering the lack of Os isotopic perturbations around OAE1d, this mercury enrichment is probably more related to local perturbation with limited influence on global climate. Major Cretaceous OAEs (OAE1a, Wezel, Fallot, and OAE2) are represented by thick organic-rich intervals, whereas the sedimentary expression of OAE1c and OAE1d in the Umbria–Marche Basin consist of cyclic alternations of thin black shales⁸. Similar cyclic intercalations of thin black shale layers in a carbonate sequence have been observed in the Valanginian–Barremian, Albian, and upper Cenomanian in the Umbria–Marche Basin (Fig. 3)^{8,22}. During Quaternary, astronomically modulated monsoonal activity cyclically enhanced the hydrology of the Mediterranean Sea at low latitude, which supplied freshwater and nutrients to the peri-continental ocean. The resulting input of terrigenous organic matter, stratification, and slightly enhanced productivity led to oxygen-depleted bottom-water conditions and the deposition of organic-rich sediments dominated by the terrigenous origin^{54–55}. Thus, the lack of the unradiogenic Os isotopic shift and the cyclic deposition of thin black shale layers during OAE1c and OAE1d may suggest a regional-scale weak marine anoxia caused by monsoonal activity modulated by astronomical cycles as proposed by previous studies⁵⁵ rather than an episodic large volcanic event⁵³. The increase in the primary productivity was not significant at the Tethyan region during OAE1d⁵⁵. However, a small positive carbon isotopic excursion during OAE1d suggests a slight increase in the primary production (Fig. 2). In addition, organic-rich sediments are reported from the Calera Limestone in California, which was deposited in the Pacific Ocean, and thus the oxygen-depleted condition could have prevailed in the East Pacific as well¹⁴. Thus, the latter process can also cause a supra-regional increase in productivity to some extent.”*

2-14. Comment

L. 277: “please add some refs.”

Reply

L.318: We added the reference at this part.

2-15. Comment

L. 299: “Bottini & Erba 2018”

Reply

L. 355. We cited the paper here.

2-16. Comment

L. 302-303: “any ref? O'Brien et al 2017; McAnena et al. 2013; Bottini et al 2015;”

Reply

L. 359: We cited O'Brien et al (2017) and McAnena et al. (2013) here. However, because of the limited number of references, we gave up citing Bottini et al. (2015).

2-17. Comment

L. 581: “Letters in the stars are not well visible.”

Reply

Fig. 1: We made the letters in the stars more visible.

2-18. Comment

L. 599: “che time scale usata?”

Reply

L. 692. We followed the age scale of Gale et al. (2020) and add the reference.

Reviewer #3

3-1. Comment

The authors have recently published Os-isotope data from the Cretaceous of Marche Umbria (Poggio le Guaine core) covering much of the Aptian–Albian interval, and there are pre-existing data from various parts of the world for the OAE 1a and OAE 2 intervals, both of which show major shifts to relatively unradiogenic values of Os and Sr isotopes (e.g. Turgeon and Creaser, Du Vivier et al., Jones and Jenkyns, Tejada et al., Bottini et al.: well reviewed in Dickson et al in Large Igneous Provinces: A Driver of Global Environmental and Biotic Changes, Geophysical Monograph 255). The authors put all this together with some new data from the Cenomanian of the Bottaccione Gorge near Gubbio and an ODP Site on the Exmouth Plateau off Australia to create some relatively

long chemostratigraphic time series, in which they can identify a number of Os-isotope excursions of likely global significance. The authors discuss the putative roles of volcanism and hydrothermal activity in governing the Os-isotope signatures of seawater and endorse the currently accepted interpretations in the literature. Because most people have focused on 'events' in their search for geochemical anomalies, the background has tended to be ignored (and may yet produce some surprises), so more extensive time series are to be welcomed. However, I'm not sure whether the simple two-fold division of OAEs into volcanic- and monsoon-related holds up to scrutiny (see below). Further, the authors have ignored some key literature that goes against their argumentation, thereby weakening the impact of the paper.

Reply

Thank you for the valuable comments. Following your advice, we added a detailed explanation of the classification of the mid-Cretaceous OAEs.

Firstly, we have added the explanation of the OAE1b because it is composed of different types of black shale layers: (L.342-350) "*Among the mid-Cretaceous OAEs, OAE1b is a problematic example. In the Umbria–Marche Basin OAE1b is composed of several major organic-rich horizons (Jacob, Kilian, Urbino, and Leenhardt Levels) intercalated with numerous thin black shale horizons¹². Although the short unradiogenic Os isotopic shifts have been reported around the Kilian Level, other black shale horizons lack unradiogenic Os isotopic shifts¹⁶. Besides, the upper part of the OAE1b is characterized by the temporal radiogenic Os isotopic excursions, which is a different feature from other mid-Cretaceous OAEs. Since OAE1b continued for several million years and contains different types of organic-rich sediments, we considered that OAE1b may be a mixture of volcanic- and monsoon-induced OAEs.*".

Besides, we added important references which seem to contradict our arguments and explanations on the points. For example, OAE1d was widespread and organic-rich sediments have been reported in the Pacific region. Thus, we added explanations on this point as (L. 318-324) "*The increase in the primary productivity was not significant at the Tethyan region during OAE1d⁵⁵. However, a small positive carbon isotopic excursion during OAE1d suggests a slight increase in the primary production (Fig. 2). In addition, organic-rich sediments are reported from the Calera Limestone in California, which was deposited in the Pacific Ocean, and thus the oxygen-depleted condition could have prevailed in the East Pacific as well¹⁴. Thus, the latter process can also cause a supra-regional increase in productivity to some extent.*".

We have also added some explanations on the sulfur isotopic fluctuations during OAE2 because it seems concordant to our hypothesis (L. 262-268): " $\delta^{34}\text{S}_{\text{barite}}$ data around OAE2

are scarce, but $\delta^{34}\text{S}$ of pyrite and carbonate-associated sulfates (CAS) around OAE2⁵¹ has been intensively investigated instead. $\delta^{34}\text{S}_{\text{CAS}}$ and $\delta^{34}\text{S}_{\text{pyrite}}$ showed a positive excursion (2–4‰) across the OAE2, suggesting an enhanced sulfate reduction⁵¹. Considering the global oceanic anoxia and short duration of unradiogenic Os isotopic shift during OAE2 (~600 kyr), the effect of the sulfate reduction could have overwhelmed the effect of volcanic sulfur input."

Finally, we mentioned the paper discussing the volcanic events during OAE1d based on mercury as following (L. 298-304) "*Therefore, we consider that the onsets of OAE1c and OAE1d were unrelated to massive submarine volcanism, unlike other major mid-Cretaceous OAEs. Mercury anomaly has been reported just below the OAE1d horizon at the Youxia section, the eastern Tethys, which has been interpreted as the submarine volcanic eruption at the Kerguelen Plateau*⁵³. *However, considering the lack of Os isotopic perturbations around OAE1d, this mercury enrichment is probably more related to local perturbation with limited influence on global climate.*"

3-2. Comment

Line 44: 'palaeo-' or 'paleo-'? Be consistent! ('palaeo-') in British English

Reply

L. 45: We changed "paleo-" to "palaeo-".

3-3. Comment

Line 45: TEX₈₆ data from the uppermost Cretaceous of the Arctic Ocean suggest sea-surface temperatures in excess of 15°C.

Reply

Thank you for the comment. We checked the TEX₈₆ temperature and found the article of Jenkyns et al. *Nature* (2004) discussing the Maastrichtian arctic temperature. We would like to mention the climatic condition of the mid-Cretaceous in the Introduction section. However, Jenkyns et al. (2004) mentioned only Maastrichtian, therefore, we decided not to cite it.

3-4. Comment

Line 57: Robinson et al. recognize an OAE1d black shale in the Calera Limestone of California, which derives from the paleo-Pacific (Geological Society of America Bulletin, 120, pp.1416-1426). Hence, the reach of this event may be greater than just Tethys/Atlantic. The Pacific also has a record of OAE 1a and OAE 2 black shales that reflect major events. How does this Calera Limestone occurrence affect the interpretation

of OAE 1d as monsoon-related? Note also the new mercury evidence bearing on OAE 1d: Yao, H., Chen, X., Yin, R., Grasby, S.E., Weissert, H., Gu, X. and Wang, C., 2021. Mercury evidence of intense volcanism preceded oceanic anoxic event 1d. *Geophysical Research Letters*, 48, p.e2020GL091508. Some reconsideration is warranted here, even if the absence of an Os-isotope signal may be telling us something.

Reply

Thank you for the constructive comments. We have added an explanation of the Pacific record of OAE1d in the Introduction as (L. 57 to 59) “*Additionally, other minor OAEs (e.g., OAE1b, OAE1c, and OAE1d), which have been reported mainly from the Tethys and Atlantic Oceans^{9,11-13} and a part of the Pacific region¹⁴, are regarded as regional to supra-regional marine anoxic events.*” and discussion section as (L.318-324) “*The increase in the primary productivity was not significant at the Tethyan region during OAE1d⁵⁵. However, a small positive carbon isotopic excursion during OAE1d suggests a slight increase in the primary production (Fig. 2). In addition, organic-rich sediments are reported from the Calera Limestone in California, which was deposited in the Pacific Ocean, and thus the oxygen-depleted condition could have prevailed in the East Pacific as well¹⁴. Thus, the latter process can also cause a supra-regional increase in productivity to some extent.*”.

Besides, we added the discussion of mercury anomaly in the main text. Considering the studied site was located near the Kerguelen Plateau, this mercury anomaly could represent the local small volcanic events at the Kerguelen Plateau. We added the explanation as following (L. 299–304) “*Mercury anomaly has been reported just below the OAE1d horizon at the Youxia section, the eastern Tethys, which has been interpreted as the submarine volcanic eruption at the Kerguelen Plateau⁵³. However, considering the lack of Os isotopic perturbations around OAE1d, this mercury enrichment is probably more related to local perturbation with limited influence on global climate.*”.

3-5. Comment

Line 104: I agree that considering OAE 1b as a cluster of events is the most useful approach. The Kilian is now taken to mark the Aptian-Albian boundary (GSSP: see Kennedy et al., *Episodes*, vol. 40).

Reply

Thank you for the comments. We tried to add the explanation and citation on the Aptian–Albian boundary at this point. However, the description of the just AAB seems not important in our paper. Besides, the number of references is limited, we gave up citing the reference here.

3-6. Comment

Line 107: it is not clear to me what ‘partially records OAE 1c’ means. How is (or was) OAE 1c defined? Maybe go back to Arthur, M.A., et al., 1990. Stratigraphy, geochemistry, and paleoceanography of organic carbon-rich Cretaceous sequences. In Cretaceous resources, events and rhythms (pp. 75-119). – who suggested the OAE nomenclature.

Reply

We modified the explanation and reference of Amadeus segment and OAE1c in the main text as (L.111-114) “*A peculiar ~2-m-thick interval in the upper Albian, called the Amadeus segment²⁴, is located in the middle part of OAE1c that spans almost the entire Biticinella breggiensis planktonic foraminiferal Zone^{8,25}.*”.

3-7. Comment

Line 132: Reference 9 does not discuss carbon-isotope data. Refer to the paper of Gambacorta et al. (Newsletters on Stratigraphy, 48, 299-323) for carbon-isotope patterns in the Upper Cretaceous of Marche–Umbria.

Reply

L.138. We added the reference here.

3-8. Comment

Line 184: I could not find a reference to modelling of weathering of Ontong Java in Tejada et al., although they suggest various types of basalt–seawater interaction to produce the Os-isotope excursion . Presumably warm- to low-temperature submarine weathering of basalt could be part of this.

Reply

Thank you for the comment. As you have indicated, the hydrothermal activity includes the weathering under the submarine condition though the estimation of its effect is unclear at present. We modified the discussion as (L. 174-179) “*these unradiogenic Os isotopic shifts correspond to the radiometric ages of the Ontong Java, Manihiki, and Hikurangi Plateaus, which once formed a single large oceanic plateau called “Ontong Java Nui” (OJN) (Fig. 3a, e), these unradiogenic Os isotopic shifts were likely triggered by a massive input of mantle-derived unradiogenic Os through hydrothermal activity and warm- and low-temperature submarine weathering at OJN^{13,20,21,34}.*”.

3-9. Comment

Line 189: during the early Albian

Reply

Line 194: We modified here to “during the early Albian”

3-10. Comment

Line 192: belemnites (plural)

Reply

Line 197: we changed “belemnite” to “belemnites”

3-11. Comment

Line 199:supports

Reply

Line 204: We changed “support” to “supports”.

3-12. Comment

Line 217: in what way is the geochemical behaviour of Os different from that of Sr in this context?

Reply

When Os was oxidized, it is volatile and easy to move. This effect during the LIPs eruption is a very critical factor to determine the Os behavior during the eruption. We added the explanation in the main text as (L. 222) “*and the volatile feature of highly oxidized form of OsO₄*”.

3-13. Comment

Line 228: presumably you exclude the possibility that the positive sulphur-isotope excursion was due to increased pyrite burial in black shales?? As illustrated in Fig. 3c, most of the increase in S-isotope values took place in the Cenomanian, post-dating OAE 1d; in so doing, S isotopes rise in parallel with established carbon-isotope curves (e.g. Jarvis et al. for the English Chalk) likely signifying gradually increasing global marine carbon burial in the run-up to OAE 2. The alternative to the idea of a decrease in hydrothermal input of sulfur, namely that the data reflect synchronous increased burial of isotopically light carbon and sulphur, needs to be discussed.

Reply

Thank you for the comment. We checked the carbon isotopic excursions during Cenomanian-Turonian at the English Chalk (Jarvis et al., 2006 Geological Magazine). The onset of the positive carbon isotopic excursion occurs after the positive sulfur isotopic excursion at the English Chalk. Since the residence time of sulfur is much longer than

carbon, the fact that carbon isotopic excursion postdates the sulfur isotopic ratio is hard to explain. Besides, in the Umbria-Marche Basin, abundant organic-rich sediments are observed during the Albian and the latest Cenomanian after the mid-Cenomanian event, which does not match the timing of sulfur isotopic excursion. Therefore, we consider that the cease in the volcanic sulfur is the most likely candidate for triggering the positive sulfur isotopic excursion as suggested in Laakso et al. (2020). The time lag between Os and sulfur isotopic ratio is derived from the large differences in the residence time. Because of the extremely long residence time of sulfur compared to Os, the sulfur isotopic ratio may have postdated the Os isotopic curve. Also, the sulfur isotopic data during Albian is scarce, and more detailed data will be required for further discussions on this point. We have explained and discussed in the main text as following (L.235-244) “*The positive excursion of $\delta^{34}S_{barite}$ can be also explained by an increase in sulfur reduction during the early Cenomanian. However, considering the organic-rich sediments are more pronounced during Albian than Cenomanian at the Umbria–Marche Basin, sulfate reduction should also have been more significant during Albian than Cenomanian. Therefore, we consider that the decrease in the volcanic sulfur is the more important factor for the positive $\delta^{34}S$ excursion rather than the sulfate reduction. The positive excursion of $\delta^{34}S_{barite}$ during the Cenomanian postdates the cease of the Os isotopic fluctuations (Fig. 3a,c). Since the residence time of sulfur in the ocean is longer than Os, the onset of the changes of $\delta^{34}S_{barite}$ could have been more gradual and possibly postdated the radiogenic Os isotopic shift.*”

3-14. Comment

Line 244: Sulfur-isotope evolution around OAE 2 (data from the English Chalk and the Scaglia in Marche–Umbria) is discussed in detail in papers by Jeremy Owens et al. (PNAS, 2013 and Sedimentology, 2017) and Jenkyns et al. (Sedimentology, 2017): the pattern is complex, with relatively high values post-OAE 2 recorded in both pyrite and CAS – exactly the opposite of what is described in the text by the authors. The Paytan et al. reference given is inappropriate.

Reply

Thank you for the comment. Several papers document the sulfur isotopic data of CAS and pyrite has been reported on the OAE2 (e.g., Ohkouchi et al., Owens et al., 2013); they suggest the positive excursion across the OAE2 which has been interpreted as the enhanced sulfate reduction and burial of sulfur with negative $\delta^{34}S$ values. Since $\delta^{34}S_{barite}$ data are scarce around OAE2, we added the citation of Owens et al. (2013) and added the detailed discussion on the $\delta^{34}S$ focusing on OAE2 as (L.262-268) “ $\delta^{34}S_{barite}$

data around OAE2 are scarce, but $\delta^{34}\text{S}$ of pyrite and carbonate-associated sulfates (CAS) around OAE2⁵¹ has been intensively investigated instead. $\delta^{34}\text{S}_{\text{CAS}}$ and $\delta^{34}\text{S}_{\text{pyrite}}$ showed a positive excursion (2–4‰) across the OAE2, suggesting an enhanced sulfate reduction⁵¹. Considering the global oceanic anoxia and short duration of unradiogenic Os isotopic shift during OAE2 (~600 kyr), the effect of the sulfate reduction could have overwhelmed the effect of volcanic sulfur input.”

3-15. Comment

Line 248: Most people would not consider the Wezel and Fallot Events as major OAEs. How extensive is their sedimentary record outside Europe?

Reply

Thank you for the comment. Fallot and Wezel Levels are only reported from the Tethys region and their precise extent is not constrained so far because of the limited geological information covering the interval outside the Tethyan region. Thus, we added this explanation in the main text as (L. 105-108) “*The Fallot and Wezel Levels are only reported from the Tethyan Region. However, their accurate extent has not been constrained so far because of the limited geological research focusing on their equivalent intervals outside the Tethyan region.*” and modified the expressions as (L. 270-272) “*Previous studies have revealed the onsets of the major Cretaceous OAEs (OAE1a, Wezel, Fallot, and OAE2) in the Tethyan region correspond to unradiogenic Os isotopic shifts^{13,16,19–21}.*”

3-16. Comment

Line 249: ‘is compatible with’ rather than ‘suggests’.

Association does not prove cause and effect.

Reply

L. 272. We modified “*suggesting*” to “*which is compatible with*” following your advice.

3-17. Comment

Lines 249–256: This paragraph is presented as if it is novel – but the ideas presented have been in the literature for many years.

Reply

L. 270-272. Thank you for the comment. We modified the sentence to avoid misleading as “*Previous studies have revealed the onsets of the major Cretaceous OAEs in the Tethyan region (OAE1a, Wezel, Fallot, and OAE2) correspond to unradiogenic Os isotopic shifts^{13,16,19–21}.*”

3-18. Comment

Lines 262–263 see comments above as to evidence for volcanic activity around OAE 1d. See also: Wilson, P.A. and Norris, R.D., 2001. Warm tropical ocean surface and global anoxia during the mid-Cretaceous period. *Nature*, 412(6845), pp.425-429 – who provide documentary evidence for the global reach of OAE 1d, extending into the Pacific.

Reply

Thank you for the suggestion. We have added the discussion on the volcanic activity during OAE1d as (L. 299–304) “*Mercury anomaly has been reported just below the OAE1d horizon at the Youxia section, the eastern Tethys, which has been interpreted as the submarine volcanic eruption at the Kerguelen Plateau⁵³. However, considering the lack of Os isotopic perturbations around OAE1d, this mercury enrichment is probably more related to local perturbation with limited influence on global climate.*”.

We have checked the TOC data and biostratigraphy and recovery of the Pacific DSDP sites (167, 463, 465, and 466) described in Wilson and Norris (2001). Although the upper Albian of Sites 465 and 466 were dark-colored sediments with relatively high TOC, the recovery of the core and their age determination is too poor to conclude that they are the OAE1d equivalent interval. Besides, we could not find the pronounced organic-rich sediments in the late Albian of DSDP Site 167 and 463. Thus, instead of the suggested paper, we mentioned the article describing the Calera Limestone as evidence of Pacific OAE1d which is a biostratigraphically well-dated section as (L. 318-324) “*The increase in the primary productivity was not significant at the Tethyan region during OAE1d⁵⁵. However, a small positive carbon isotopic excursion during OAE1d suggests a slight increase in the primary production (Fig. 2). In addition, organic-rich sediments are reported from the Calera Limestone in California, which was deposited in the Pacific Ocean, and thus the oxygen-depleted condition could have prevailed in the East Pacific as well¹⁴. Thus, the latter process can also cause a supra-regional increase in productivity to some extent.*”.

3-19. Comment

Lines 316–317: what about the Caribbean Plateau?

Reply

Thank you for the comment. We have added the description of the Caribbean Plateau as (L. 377-379) “*At the Cenomanian–Turonian boundary, the massive submarine volcanic eruptions at the Caribbean Plateau could have been further contributed to the enhancement of global warming during the OAE2⁶².*”.

3-20. Comment

Line 374: change 'The since' to 'Sincere'

Reply

Line 640. We modified "since" to "sincere".

3-21. Comment

References: many abbreviations are not complete and/or not compatible with journal style. Special Publications/Papers (e.g. Fassell and Bralower; Savian et al.) are not clearly distinguished. Other examples:

Reply

Thank you for the comments. We modified the abbreviations of the journal titles (L.439-637).

3-22. Comment

Line 526: reference should begin: Pogge von Strandmann, P.A.E. (see also line 488)

Reply

L.537. Thank you for the comment. We have modified this reference.

3-23. Comment

Line 546: reference incomplete

Reply

Thank you for the comments. We have modified this reference (L. 599).

Reviewer #2

Comment

The authors have provided an improved version of their work which takes into account the suggestions of the reviewers. I have especially found improvements in the discussion which are now more complete. I do recommend the publication.

In the current version I spotted a few minor things reported as follows:

Reply

We warmly thank the reviewer for having appreciated the efforts in improving the manuscript and recommending its acceptance.

Comments

2-1. Comment

lines 165 and 167 “represent” is repeated twice.

Reply

L. 167. We change “represent” to “reflect”.

2-2. Comment

line 240 “is the more” I do not think this is correct in english, please check

Reply

Thank you for your comments. We rephrased this part as “*Therefore, we consider that the decrease in the volcanic sulfur is a more important factor for explaining the positive $\delta^{34}\text{S}$ excursion than the sulfate reduction.*” (L. 239–241).

2-3. Comment

line 270 “Previous studies have revealed THAT the onsets of the major Cretaceous OAEs (OAE1a, Wezel, Fallot, and OAE2) in the Tethyan region correspond to....”

Reply

L. 270. We added “that” before “the onset”.

2-4. Comment

Fig. 3 “Furlo” is misspelled and box “e” and “f” are not aligned

Reply

We corrected the spelling in Fig. 3 and revised boxes (e) and (f).

Reviewer #3

Comment

The authors have addressed the most significant points raised in the reviews. I note some minor points in the text that need attention and some more substantive issues.

Reply

We warmly thank the reviewer for having appreciated the efforts in improving the manuscript and raised further minor comments that certainly have improved it.

3-1. Comment

Line 242: change 'cease' to 'cessation'

Reply

L.242. We changed “cease” to “ cessation”.

3-2. Comment

Line 273: change to 'volcanism and OAEs.'

Reply

L. 273. We changed this part following your suggestion.

3-3. Comment

Line 281: change 'mental' to 'metal'

Reply

L. 281. we changed 'mental' to 'metal'

3-4. Comment

Line 299: change to 'A mercury anomaly . . .'

Reply

L. 299. We inserted “A” before mercury.

3-5. Comment

Line 309: change to 'During the Quaternary . . .'

Reply

L. 309. We inserted “the” before “Quaternary”.

3-6. Comment

Line 381: look at: Scaife, J.D. et al., 2017. Sedimentary mercury enrichments as a marker for submarine large igneous province volcanism? Evidence from the Mid-Cenomanian event and Oceanic Anoxic Event 2 (Late Cretaceous). *Geochemistry, Geophysics, Geosystems*, 18, 4253–4275.

The mercury evidence is instructive in terms of proposed submarine/subaerial LIPs.

Reply

Thank you kindly for letting us know about the interesting and important paper. The mercury evidence described in this paper may be consistent with the idea that the volcanic event at the Caribbean Plateau occurred under submarine conditions and could not have contributed emission of greenhouse gasses for a long-time range. However, we feel that it may be necessary to further discuss the transportation process of mercury using various proxies and modeling. Besides, the number of citations of this article already surpasses the maximum numbers. For these reasons, we gave up including it in this article.

3-7. Comment

Page 21: in their discussion of Cenomanian–Turonian temperatures the authors state that no long-term evidence for hydrothermal activity is recorded and they reference Fig. 3. However, there is very little in the way of an illustrated Turonian record in this figure and notably marine Sr-isotope keep falling from the OAE 2 interval (big shift) to less radiogenic values over the whole of the Turonian (see compilation in Jones and Jenkyns (*American Journal of Science*) and papers by McArthur et al. Maybe the Caribbean, High Arctic and other LIPS were supplying non-radiogenic Sr (and Os?) over this period?

Reply

Thank you for the comments. As it was suggested by the reviewer, there have been no long-term Os isotopic data during Turonian. Thus, it is not appropriate to describe as “*Furthermore, the highest temperature during the mid-Cretaceous was recorded during the Cenomanian–Turonian, from which no long-term hydrothermal activity associated with LIPs formation has been reported*”. Therefore, we rephrase this part as (L.360–362) “*Furthermore, the highest temperature during the mid-Cretaceous was recorded during the Cenomanian–Turonian. However, no long-term hydrothermal activity associated with LIPs formation has been reported during the Cenomanian (Fig. 3).*”.

During the Turonian, Sr isotope ratio showed an unradiogenic shift. The interval corresponds to the volcanic events at the Caribbean Plateau, High Arctic LIPs, Madagascar Plateau, and circum-Pacific volcanism. Especially, submarine volcanic events at the Caribbean and Madagascar Plateaus could have been the cause of the unradiogenic Sr isotopic shift. However, since no Os data in Turonian are available, this

interval will be our next target.

Figure 3: the two right-hand columns are offset from the other three. Should they not line up?

Reply

Figure 3. We modified Figure 3.

REVIEWERS' COMMENTS

Reviewer #3 (Remarks to the Author):

I think that the paper is now broadly acceptable. I note a few points, mostly concerning grammar and style:

Line 56: the Bonarelli Level is only about a metre thick; the Selli Level somewhat thicker. But 'thick' is a relative term and some people may be thinking in terms of tens of metres. Better to say 'metre-thick organic-rich horizons'.

Line 109: there is a mixture of French and Italian names here. Remember that Urbino corresponds to the Paquier. The French names all come from the Vocontian Trough in SE France

Line 197: briefly explain (in brackets) that glendonites are deposits formed after the low-temperature form of hydrated calcium carbonate known as ikaite.

Line 208: 'in the Indian realm'

Line 239: 'during the Albian and the Cenomanian'

Line 245: 'that also support'

Line 258: the Du Vivier et al., paper (EPSL) really should be cited here as they specifically discuss the Caribbean LIP

Line 288/289: the hyphenation is not correct

Line 291; 'expanded' would be a better word than 'thickened'

Line 295/196: 'Amadeus Segment'

Line 303: 'perturbations' would be better

Line 305: be more specific over thicknesses

Line 314: 'dominated by terrigenous sources' would be better

Line 319: 'in the Tethyan region'

Line 348: change to 'which constitute a different feature'

Line 405: change to 'corresponded' Geological narrative is past tense

Reviewer #3 (Remarks to the Author):

I think that the paper is now broadly acceptable. I note a few points, mostly concerning grammar and style:

We would like to sincerely thank the reviewer for many constructive comments and hope that our revised manuscript will meet the reviewer's comments.

Comments 3-1.

Line 56: the Bonarelli Level is only about a metre thick; the Selli Level somewhat thicker. But 'thick' is a relative term and some people may be thinking in terms of tens of metres. Better to say 'metre-thick organic-rich horizons'.

Reply

L. 56–57. We corrected this part as “*were typified by worldwide deposition of thick organic-rich horizons (e.g., 1–2 meter-thick at the Umbria–Marche Basin)*”.

Comments 3-2.

Line 109: there is a mixture of French and Italian names here. Remember that Urbino corresponds to the Paquier. The French names all come from the Vocontian Trough in SE France

Reply

L. 122. Thank you for your comment. We changed “Urbino Level” into “*Urbino/Paquier Level*”.

Comments 3-3.

Line 197: briefly explain (in brackets) that glendonites are deposits formed after the low-temperature form of hydrated calcium carbonate known as ikaite.

Reply

L. 211–213. We added the explanation of glendonite as “*(a pseudomorph after ikaite that is a hydrated calcium carbonate formed under low-temperature conditions)*”

Comment 3-4.

Line 208: 'in the Indian realm'

Reply

L. 223. We changed “at the Indian realm” into “*in the Indian realm*”.

Comment 3-5.

Line 239: 'during the Albian and the Cenomanian'

Reply

L. 254. We added “the” before “*Albian*”.

Comment 3-6.

Line 245: 'that also support'

Reply

L. 260. We changed “supports” into “support”.

Comment 3-7.

Line 258: the Du Vivier et al., paper (EPSL) really should be cited here as they specifically discuss the Caribbean LIP

Reply

L. 270 We cited Du Vivier et al. (2014) at this part.

Comment 3-8.

Line 288/289: the hyphenation is not correct

Reply

L. 305–306 We modified this part as “*deep-, intermediate-, and surface- ocean*”

Comment 3-9.

Line 291; 'expanded' would be a better word than 'thickened'

Reply

L. 308 We changed “thickened” into “expanded”.

Comment 3-10.

Line 295/196: 'Amadeus Segment'

Reply

L. 297 We changed “Amadeus segment” into “Amadeus Segment”.

Comment 3-11.

Line 303: 'perturbations' would be better

Reply

L. 320 We changed “perturbation” into “perturbations”.

Comment 3-12.

Line 305: be more specific over thicknesses

Reply

L. 322 We added specific thicknesses as “~6 cm to 2 m”.

Comment 3-13.

Line 314: 'dominated by terrigenous sources' would be better

Reply

L. 331 We changed “the terrigenous origin” to “terrigenous sources”

Comment 3-14.

Line 319: 'in the Tethyan region'

Reply

L. 336 We changed “at” into “in”.

Comment 3-15.

Line 348: change to 'which constitute a different feature'

Reply

L. 365 We changed “which is a different feature” into “*which constitute a different feature*”.

Comment 3-16.

Line 405: change to 'corresponded' Geological narrative is past tense

Reply

L. 416 We changed “corresponds” to “corresponded”.